# Topological Classification of Correlations in 2D Electron Systems in Magnetic or Berry Fields

**DOI:** 10.3390/ma14071650

**Published:** 2021-03-27

**Authors:** Janusz E. Jacak

**Affiliations:** Department of Quantum Technologies, Wrocław University of Science and Technology, Wyb. Wyspiańskiego 27, 50-370 Wrocław, Poland; janusz.jacak@pwr.edu.pl

**Keywords:** homotopy phases, long-range quantum entanglement, FQHE, Hall systems, Chern topological insulators

## Abstract

Recent topology classification of 2D electron states induced by different homotopy classes of mappings of the planar Brillouin zone into Bloch space can be supplemented by a homotopy classification of various phases of multi-electron homotopy patterns induced by Coulomb interaction between electrons. The general classification of such type is presented. It explains the topologically protected correlations responsible for integer and fractional Hall effects in 2D multi-electron systems in the presence of perpendicular quantizing magnetic field or Berry field, the latter in topological Chern insulators. The long-range quantum entanglement is essential for homotopy correlated phases in contrast to local binary entanglement for conventional phases with local order parameters. The classification of homotopy long-range correlated phases induced by the Coulomb interaction of electrons has been derived in terms of homotopy invariants and illustrated by experimental observations in GaAs 2DES, graphene monolayer, and bilayer and in Chern topological insulators. The homotopy phases are demonstrated to be topologically protected and immune to the local crystal field, local disorder, and variation of the electron interaction strength. The nonzero interaction between electrons is shown, however, to be essential for the definition of the homotopy invariants, which disappear in gaseous systems.

## 1. Introduction

The discovery of fractional quantum Hall effect in 2D electron systems (1982, experiment [1], 1983, theory [2]) followed former observation of the integer quantum Hall effect for 2D electrons (1980 [3]) pointed out the role of interaction of electrons in organization of strongly correlated multiparticle systems in planar geometry. Earlier, the special topological behavior in 2D electron systems was proposed [4] to remedy the disruption of long range order in planar systems [5]. In 2D electron systems, phase transitions with broken symmetry and with local order parameter cannot occur because a long range order is destabilized by planar Goldstone excitations [6]. In addition, in quantum Hall states in 2D, no local order parameter exists and any symmetry is broken. Instead of this, the topological multi-electron phases occur with various long range multi-particle correlations protected by homotopy invariants. The experimental and theoretical study of such collective planar states flourished after the discovery of graphene and the development of Hall-type experiment in monolayer and bilayer graphene, which achieved unattainable previously precision. The specific 2D quasi-relativistic dynamics in graphene [7] connects with the former idea of quantum Hall-type behavior without Landau levels [8], generalized next onto a wide family of materials called topological insulators [9]. The predominant factor unifying all these phenomena is the planar geometry of the physical space where electrons are located, which opens an avenue to topological effects both in single particle planar dynamics and in collective multi-electron planar-correlation effects conditioned by electron interaction. To describe and classify such topological correlations, the homotopy methods of algebraic topology are especially convenient [10,11].

To classify general topological properties, the notion of invariants for continuous transformations of topological spaces is utilized. These invariants, if used in physical systems, are robust against local disorder and thermal chaos and are thus suitable to characterize topologically protect stable collective states. The most important invariants are homotopy groups linked to homology and cohomology [10,11]. The algebraic structure of homotopy groups defined on some space reflects the global properties of this space in terms of continuity of mappings, which describe trajectories or surfaces and hyper-surfaces in this space. The homotopy groups are named as $\pi_{i}$ groups. For i=1, the $\pi_{1}\left(A\right)$ group (the first homotopy group, frequently called also as the fundamental group of *A*) collects disjoint classes of trajectories (closed loops) in the space *A*, which cannot be transformed one into another by any continuous deformation without cutting (such trajectories are called nonhomotopic). The classes of non-equivalent up to continuous deformations surfaces build $\pi_{2}\left(A\right)$ group and $\pi_{i}\left(A\right)$, i>2 for hyper-surfaces of higher dimension [10]. This mathematical apparatus finds applications in physics: in crystallography to classify defects, in condensed matter to classify textures in various phases with multi-component order parameters like in liquid crystals or in superfluid He${}^{3}$ [10], in field theory to classify instantons [12] and, recently, to classify 2D topological insulators, 2D superconductors, and quantum Hall phases [13,14,15]. In particular, in the topological insulator case, the topological invariants may numerate various nonhomotopic mappings from the 2D Brillouin zone into the Bloch space and in this way they may distinguish between different types of band structures topologically protected and typically linked with the presence of Dirac-like points known from graphene and topological insulators [7,9]. However, the band structure is a single-particle quantum problem not concerning mutual interaction of electrons.

The multi-particulate correlation patterns that may manifest macroscopically are the matter of electron interaction and they occur particularly spectacularly in 2D multi-electron systems. Topological invariants appear to be helpful for the characterization and classification of the multi-particle correlation collective effects driven by the electron interaction which are beyond the single-particle band structure restrictions (like in topological insulators) or conventional phase transitions due to spontaneous breaking of some symmetry and coherent but binary channels of electron scattering (e.g., in superconductors) or binary spin interaction (in magnetic phases) [16]. Topological correlations and related quantum homotopy phase transitions [17] differ from the conventional thermodynamic phase transitions with local order parameter and also local binary quantum entanglement due to some selected binary channel of electron interaction. Topological homotopy phases are not linked with any local order parameters but display the correlation patterns between all electrons simultaneously and thus exhibit the long-range quantum entanglement of all electrons in the system [18].

In the present paper, we provide a homotopy classification of correlation patterns of 2D interacting multi-electron planar systems at the perpendicular magnetic field, utilizing the notion of the cyclotron commensurability of electrons with 2D periodic lattice analogous to the previously discussed problem of the fine structure of Landau levels (LLs) induced by a periodic crystal potential referred to as a Hofstadter butterfly [19,20]. The fine fractal structure of LLs of 2D electrons induced by the external crystalline 2D lattice has been demonstrated in these papers and the graphical presentation of the result resembles a butterfly shape (as illustrated in Figure 1).

The periodic crystal potential was in this study external, so that the problem was of the single-electron type. In the present paper, we propose to consider the cyclotron commensurability of 2D electrons with the Wigner crystal of electrons themselves instead of an external crystal lattice. This makes the problem multi-electron and essentially collective, interaction dependent as opposed to single-electron non-collective problem of fine LL structure in periodic external 2D potential at the Hofstadter butterfly effect. The interaction of electrons will now play a fundamental role because the organization of the Wigner crystal is a matter of the electron repulsion; thus, the problem is of a multi-particle type. To distinguish between various classes of 2D electron correlations induced by their Coulomb interaction and defined by the cyclotron commensurability of electron dynamics with the electron Wigner lattice, the homotopy approach is especially convenient and allows for the identification of topological invariants, which protect different types of multi-electron correlations.

The present consideration is a continuation of our former result related to a Feynman path integral in multiple-connected coordination space of 2D particles [17]. The novelty in the present paper is the discussion of the symmetry of multi-particle wave functions independent of path integral in the way suggests by Sudarshan et al. [21,22]. We demonstrate that homotopy invariants determine topologically protected multi-electron states with distinct multi-particle wave functions and energies. Moreover, the linkage of factors in the topological invariants with successive generations of the next-nearest neighbors in electron Wigner lattice is clarified for the first time. The transparent explicit example for two electrons satisfying distinct commensurability conditions resulting in different homotopy patterns is presented, which can next be generalized onto large systems of *N* electrons (in thermodynamic limits as well). The tabular summarizing of the general homotopy classification in various systems depending on material and space dimension is also presented. The paper is organized as follows. In the next paragraph, the idea of the cyclotron commensurability of electrons on the plane is sketched, and the metrics of homotopy objects are defined. These metrics are used, in the following paragraph, to classify homotopy quantum phases of 2D correlated electron systems. The close linkage of the homotopy correlation with the long range quantum entanglement [18] is demonstrated, which essentially distinguishes the topological correlated states from conventional collective phases with local order parameter and only binary quantum entanglement. Next, the developed homotopy approach is presented in application to quantum Hall physics in 2D semiconductor systems and in graphene, including graphene monolayer and bilayer. If a magnetic field is substituted by the Berry field [9], the similar homotopy classification of 2D electron correlations is extended onto Chern topological insulators.

## 2. Metrics in Full Braid Group for 2D Electrons in Magnetic Fields

Multi-particle correlated systems can be in general described in terms of the first homotopy group $\pi_{1}$ of *N*-particle configuration space, called in this case the braid group (because trajectories in multi-particle configuration space are the intertwined bundles of individual electron trajectories) [10,23]. The $\pi_{1}$ group consists of disjoint nonhomotopic trajectory classes. The trajectories (closed loops) from distinct classes cannot be transformed one onto another by any continuous deformation without cutting, and such trajectories are called nonhomotopic [11,24]. For multi-particle systems of identical and indistinguishable particles, the configuration space has the form, $F_{N}=\left(M^{N}-\Delta \right)/S_{N}$, where *M* is the physical space where particles are located (e.g., 3D space or 2D plane), $M^{N}=M\times M\times \cdots \times M$ is *N*-fold product of *M*, *N* is the number of particles (electrons) in the system, Δ is the diagonal subset of $M^{N}$ collecting points with coordinates of at least two particles coinciding and subtracted from $M^{N}$ to assure particle number conservation. The division by the permutation group $S_{N}$ introduces the indistinguishability of identical particles. The homotopy group $\pi_{1}\left(F_{N}\right)$ is called the full braid group [10,23]. The braid group collects classes of closed trajectory loops in multi-particle space $F_{N}$ defining exchanges of particles on *M*, i.e., multi-strand trajectories with start and final points which can differ by renumbering of particles only (but due to the indistinguishability of particles in the definition of $F_{N}$, these points in $F_{N}$ coincide). Distinct classes of trajectories are topologically disjoint and the trajectories from different classes cannot be transformed one into another by any continuous deformation without cutting—they are nonhomotopic.

Because the braids from the full braid group correspond to various exchanges of indistinguishable particles, thus the scalar unitary representations of braids define quantum statistics of particles. It has been proved [21,22] that any multi-particle wave function of *N* particles in the space *M* must transform according to a scalar unitary representation of the braid when the arguments of these wave functions, i.e., classical positions of particles, mutually exchange according to this particular braid [22,23]. For three-dimensional space *M*, the full braid group of *N* particles is always the finite permutation group $S_{N}$ [10,24], which only has two scalar unitary representations, corresponding to bosons and fermions. However, for two-dimensional *M*, the full braid group is infinitely complicated group [10,21,23,24,25] with many representations corresponding to anyons [23,26]. In the case of 3D space, an exchange of particle positions resolves itself to only their renumbering (permutation of indices) because the full braid group for $M=R^{3}$ is the permutation group, but in 2D the paths of exchanges of particles (electrons) are important because braids are not permutations for two-dimensional space *M* [10,23,24,25].

The simple scheme sketched above becomes, however, more complicated in the presence of a magnetic field, especially in a spectacular manner for $M=R^{2}$ and also for locally planar manifolds *M* like a sphere or torus. In the presence of the magnetic field, braids that describe exchanges of indistinguishable electrons on the plane must be built of pieces of cyclotron orbits (because no other trajectories are available when the magnetic field is switched on) and cyclotron orbits in 2D are planar without the drift motion along the field perpendicular to the plane. This influences the homotopy classes of trajectories in 2D multi-particle charged systems exposed to a perpendicular magnetic field.

The full braid group is generated by elementary braids, $\sigma_{j}$, j=1,…,N−1, being exchanges of neighboring electrons, *j*-th with (j+1)-th one at certain electron enumeration [23,24]. However, these elementary braids at magnetic field presence must be half-pieces of the cyclotron orbits, cf. Figure 2 (we remind readers that braids are classical trajectories in the multi-particle configuration space and consist of individual paths of all particles; for $\sigma_{j}$, only *j*-th and (j+1)-th particles move, while the others remain at rest [10,23]), and therefore the braids $\sigma_{j}$ are of a finite size in a 2D case, as planar cyclotron orbits are spatially ranged. Electrons in 2D at the presence of a magnetic field (neglecting interaction) fill (LLs) [27] and due to the degeneracy of these levels the cyclotron orbits for each LL are defined as $\left(2n+1\right)\frac{h}{eB}$, where $\Phi_{1}=\frac{h}{e}$ is the magnetic field flux quantum, *n* is the Landau index and *B* is the magnetic field strength. Hence, all generators $\sigma_{j}$ of the braid group have the same size for electrons in the same LL. This constant size can be considered as the metrics imposed on the braid group for these electrons. These metrics are convenient to be expressed as the surface of the cyclotron orbit (regardless of its particular shape in the interacting multi-particle system), remembering that braids $\sigma_{j}$ are half-pieces of cyclotron orbits.

The braids $\sigma_{j}$ must exchange neighboring electrons. Thus, their metrics must be precisely accommodated to electron positions in the classical electron Wigner lattice; otherwise, these generators cannot be defined. In the Wigner crystal, all electrons are uniformly distributed on the positively charged plane (jellium) and repulse themselves, thus the separation between electrons is rigidly fixed at T=0 K, cf. Figure 2. The neighboring electrons cannot be closer than this separation. Hence, the condition for the existence of the braid group with $\sigma_{j}$ generators is as follows for the lowest LL (LLL) (n=0),
(1)$$\frac{S}{N}=\frac{\Phi_{1}}{B}=\frac{h}{eB},$$
where *S* is the surface of the total sample plane, *N* is the number of electrons placed on this plane. Taking into account that the degeneracy of LLs equals $N_{0}=\frac{BSe}{h}$ (cf. Appendix A), we get from Equation (Equation 1), $\nu =\frac{N}{N_{0}}=1$, which is the filling rate for integer quantum Hall effect (IQHE). The commensurability condition (Equation 1) thus defines the correlation between all electrons in the completely filled LLL.

We see that the IQHE state is the correlated state of electrons according to the full braid group homotopy pattern (Equation 1) and this is not equivalent with only an integer filling rate of LLs (here, the LLL). Complete filling of LLs can also happen in gaseous systems without interaction, but is not protected there by any homotopy invariant in contrast to (Equation 1). The commensurability condition (Equation 1) is a matter of electron mutual interaction and cannot be fulfilled in the gas (in the gas, no Wigner lattice can be defined). Thus, we conclude that the metrics $\frac{h}{eB}$ of generators $\sigma_{j}$ together with the interaction of electrons define correlations between all electrons via the commensurability condition (Equation 1) and the corresponding multi-particle correlated quantum state is responsible for IQHE. In the gaseous system, without inter-particle interaction, no correlations exist, despite the generators $\sigma_{j}$ and their metrics still being able to be defined, but the condition (Equation 1) disappears for noninteracting particles (the metrics of braids still holds if particles are charged, but the separation of gaseous particles is arbitrary when the interaction is neglected), cf. Appendix C.

However, what about the metrics of other braid group elements? To answer this question, we will apply the Bohr–Sommerfeld (B–S) rule. The B–S rule refers quasiclassically to the case of 1D classical phase space and the trajectory of a particle in arbitrarily shaped 1D well with turning points at barriers of the well, and this rule expresses the surface in 1D phase space ranged by the phase trajectory in terms of multiplicity of its quantum, *h* (Planck constant),
(2)∮pdx=h(n+1/2).

One can apply the B–S rule to the pair of components of the kinematic momentum of an electron in 2D at the presence of perpendicular magnetic field *B*, when these components do not commute because of the field presence. Let us assume the Landau gauge for the vector potential, A=(0,Bx,0), then the kinematic momentum components have the form,
(3)$$\begin{array}{c} {\hat{P}}_{x}=-i\hbar \nabla_{x},\\ {\hat{P}}_{y}=-i\hbar \nabla_{y}-eBx, \end{array}$$
for which the commutator ${\left[{\hat{P}}_{y},{\hat{P}}_{x}\right]}_{-}=-\hbar eB$ (this commutator is gauge invariant). Therefore, one can define a pair of canonically conjugated variables, $\hat{Y}$ and ${\hat{P}}_{y}$, for which ${\left[{\hat{P}}_{y},\hat{Y}\right]}_{-}=-i\hbar$, and thus they can be treated as a generalized position, $\hat{Y}={\hat{P}}_{x}/\left(eB\right)$, and momentum, ${\hat{P}}_{y}$. From the B–S rule applied to $\hat{Y}$ and ${\hat{P}}_{y}$, we get
(4)$$\oint P_{y}dY=h\left(n+1/2\right),$$
which is the same as $\oint P_{y}dP_{x}=heB\left(n+1/2\right)$ (due to the definition of $\hat{Y}$).

The 1D phase space of these canonically conjugated variables ${\hat{P}}_{y}$ and $\hat{Y}$ coincides with the 2D space $\left(P_{x},P_{y}\right)$ if multiplied by eB. The 1D phase trajectory is thus the $\left(P_{x},P_{y}\right)$ 2D trajectory. Any trajectory in kinematic momentum space $\left(P_{x},P_{y}\right)$ is repeated in the real position 2D space (x,y), but rescaled and rotated by π/2, which follows from the Lorentz force formula, dP=−edr×B. From (Equation 4), we thus obtain for the surface of the orbit in the space (x,y),
(5)$$\oint ydx=\frac{h}{eB}\left(n+1/2\right).$$

Multiplying Equation (Equation 5) by *B*, the quantum of magnetic field flux can be determined, $\Phi_{1}=\frac{h}{e}$ resulting from Equation (Equation 5) if *n* changes by 1.

Now, let us turn back to braids in the multi-electron 2D system. Trajectories for exchanges of 2D electrons in the magnetic field presence are planar cyclotron braid trajectories. In the full braid group, we have braids which are arbitrary group products of the generators $\sigma_{j}$ (j=1,2,…,N). Especially interesting are braids $\sigma_{j}^{2k+1}$ where *k* is a positive integer. The braids $\sigma_{j}^{2k+1}$ also describe exchanges of particles *j*-th with (j+1)-th ones, but with additional *k* loops [17]. We see that braids $\sigma_{j}^{2k+1}$ repeat *k* times the exchanges of particles *j*-th and (j+1)-th ones, which were exchanged without any repetition by the generators $\sigma_{j}$. The difference between braids $\sigma_{j}^{2k+1}$ and $\sigma_{j}$ is only in additional *k* loops in the former ones. However, in the case when the generators $\sigma_{j}$ cannot be defined as too short to match the closest neighbors in the Wigner lattice (as illustrated in Figure 2c), then we anticipate that the role of the simplest braids (and generators of the braid group) is taken by $\sigma_{j}^{2k+1}$ instead of $\sigma_{i}$, provided the size (metrics) of multi-loop braids $\sigma_{j}^{2k+1}$ fits to electron separation in the Wigner lattice.

One can prove [17] that the size of braids $\sigma_{j}^{2k+1}$ actually is larger than the metrics of $\sigma_{j}$, when the latter cannot be implemented in the Wigner lattice. In the case when $\sigma_{j}$ (j=1,…,N−1) are excluded from the braid group (as too short), the simplest electron exchange trajectories can be $\sigma_{j}^{2k+1}$ and such must be taken to the B–S rule as paths back and forth in the contour integral in Equation (Equation 2) (the trajectories between turning points in the derivation of the B–S rule). Instead of Equation (Equation 2), we thus get
(6)∮pdx=(2k+1)h(n+1/2).

Repeating for Equation (Equation 6) the above presented derivation of the flux quantum $\Phi_{1}=\frac{h}{e}$, we get for multi-loop trajectories (i.e., for (2k+1)-loop cyclotron orbits, or equivalently, braids with additional *k* loops as shown in Figure 2) the effective flux quantum which equals $\Phi_{2}=\left(2k+1\right)\frac{h}{e}$—it is larger than $\Phi_{1}=\frac{h}{e}$. Larger flux quantum defines a larger size of the corresponding cyclotron orbit, $\left(2k+1\right)\frac{h}{eB}$ instead of former $\frac{h}{eB}$.

Loops can appear only one by one in the braid paths between turning points in Equation (Equation 2), and these loops are counted by k=1,2,…, thus, on the total closed phase-space cycle in the B–S rule, we have jointly 2k+1 loops and the B–S circular integral grows 2k+1 times, as shown in Equation (Equation 6).

This effective flux quantum for multi-loop orbits, $\Phi_{2}=\left(2k+1\right)\Phi_{1}$, has the larger surface, $\Phi_{2}/B$, at the same magnetic field as that for single-loop orbits with the surface, $\Phi_{1}/B$. $\Phi_{2}/B$ determines the size of the multi-loop cyclotron orbit and the metrics of corresponding braids, $\sigma_{j}^{2k+1}$. This proof holds, however, only if $\sigma_{i}$ are precluded as too short. In the case when $\sigma_{j}$ are not precluded, the braids $\sigma_{j}^{2k+1}$ have the same size (metrics) as $\sigma_{j}$ because the simplest paths between turning points in the B–S rule must be $\sigma_{j}$ and cannot be substituted by $\sigma_{j}^{2k+1}$. However, if $\sigma_{j}$ are excluded at sufficiently strong magnetic field presence, then the situation changes, as described above.

To define the braids $\sigma_{j}^{2k+1}$, the appropriate positions of the electrons in the Wigner lattice must be accessible, so that these electrons can exchange according to these braids. It is clear that such electrons must be separated in consistence with the metrics of these braids and, on the other hand, in consistence with the electron distribution in the Wigner lattice, in order to allow exchanges. The metrics of $\sigma_{j}^{2k+1}$ is larger than that of $\sigma_{j}$ (if $\sigma_{j}$ are precluded) and perfectly fits to separation of electrons $\frac{S}{N}$ at the magnetic field (2k+1)B, at which the metrics of $\sigma_{j}$ is too small, i.e., $\frac{S}{N}>\frac{\Phi_{1}}{(2k+1)B}$ (for k≥1), but $\frac{S}{N}=\frac{\Phi_{2}}{(2k+1)B}$. In such (2k+1)-times larger fields, the generators $\sigma_{j}$ cannot be defined and must be removed from the full braid group and substituted by new generators $\sigma_{j}^{2k+1}$ with larger metrics (as illustrated in Figure 2b). The new generators generate a subgroup of the full braid group (as expressed by $\sigma_{j}$) and we call this subgroup the cyclotron braid subgroup [17,28]. Again, the condition of the equality of the generator metrics with the closest particle separation (expressed as the surface per particle, $\frac{S}{N}$ with *S* and *N* kept constant) defines the homotopy invariant,
(7)$$\frac{S}{N}=\frac{\Phi_{2}}{(2k+1)B}=\frac{\left(2k+1\right)\Phi_{1}}{(2k+1)B}.$$

However, for the larger magnetic field, (2k+1)B, the degeneracy of Landau Levels grows, $N_{0}=\frac{(2k+1)BSe}{h}$, which, via Equation (Equation 7), gives $\nu =\frac{N}{N_{0}}=\frac{1}{2k+1}$. This fractional filling rates of the lowest Landau level correspond to the main Laughlin hierarchy of the fractional quantum Hall effect (FQHE) [2].

The specific phase shift of the Laughlin functions for states from this hierarchy is naturally given by the scalar unitary representation of the cyclotron braid subgroup, i.e., by the projective representation of the initial full braid group onto the cyclotron subgroup [17], $\sigma_{j}^{2k+1}\to e^{i(2k+1)\alpha }$, for original representation of the full braid group, $\sigma_{j}\to e^{i\alpha }$ (for original electrons, α=π).

Let us emphasize that the B–S rule and its application to the quantization of the magnetic field flux is independent of the interaction (as the quasiclassical approach of B–S rule is not perturbative in interaction), and thus holds for arbitrary strongly interacting systems. Hence, the metrics of multi-loop braids and related homotopy correlation patterns are invariant with respect to the interaction strength, but need the nonzero repulsion of electrons to constitute their Wigner lattice.

## 3. General Cyclotron Commensurability for 2D Interacting Electron Distribution

The commensurability condition (Equation 7) can be rewritten as follows:(8)$$\frac{S}{N}=\frac{(2k+1)h}{eB}=\frac{h}{eB}+\frac{h}{eB}+\cdots +\frac{h}{eB},$$
where the sum extends over q=2k+1 terms, i.e., over all loops of the multi-loop cyclotron orbit. The cyclotron braid commensurability scheme presented in the preceding paragraph and expressed by the above formula can be also generalized by the inclusion of the nesting of braids with next-nearest neighbors in the Wigner lattice and for each loop of the multi-loop braid separately, which leads to the commensurability condition [17,28],
(9)$$\frac{S}{N}=\frac{h}{eBx_{1}}\pm \frac{h}{eBx_{2}}\pm \cdots \pm \frac{h}{eBx_{q}},$$
where q=2k+1 is the number of cyclotron loops (*k* is the number of loops in the braid as braids in 2D are half-pieces of cyclotron orbits). The factors $x_{i}$ (i=1,…,q, q=2k+1) denote the portions of the next-nearest neighbors in the Wigner lattice selected to commensurate with the *i*-th loop of the multi-loop orbit and ± in the above formula account for congruent (+) or opposite (−) circulation of *i*-th loop with respect to the preceding one. For $x_{i}=1$ for all *i* (and taking + before each component), the formula (Equation 9) coincides with Equation (Equation 8) or Equation (Equation 7), which refer to the nearest neighbor commensurability of all loops.

To clarify the transition between Equations (Equation 8) and (Equation 9), let us first explain the meaning of $x_{i}>1$ in denominators in Formula (Equation 9). If one considers the single-loop commensurability for every *x*-th electron, i.e., for the portion $N^{\prime }$ of electrons, where $N^{\prime }=\frac{N}{x}$, then the commensurability condition attains the form, $\frac{S}{N/x}=\frac{h}{eB}$, which is the same as $\frac{S}{N}=\frac{h}{eBx}$. We see that this is of the same form as the one-loop component in the sum in Equation (Equation 9), if one takes $x_{i}=x$. Hence, the components of the sum in Equation (Equation 9) describe nesting of consecutive loops of multi-loop cyclotron orbit with next-nearest electrons and the portions of these electrons are determined by $x_{i}$ independently for each loop i=1,…,q, q=2k+1. This can be illustrated schematically in Figure 3. When all $x_{i}=1$, we arrive back at Equation (Equation 8), i.e., at nesting of all loops of the *q*-loop cyclotron orbit with only nearest neighboring electrons. To account for a most general form of the cyclotron commeasurability, we also admitted an inverse orientation of particular loop with respect to preceding one, which is marked with sign minus of ± in the sum in Equation (Equation 9).

Possible factors $x_{i}$ for electrons are precisely defined and associated with the Wigner crystal as is directly shown in Appendix B.

For each filling ratio of LLs (i.e., the specific value of magnetic field at constant *S* and *N*) when the condition (Equation 9) is satisfied, it is possible to define the cyclotron subgroup generator, ${\tilde{\sigma }}_{j}^{2k+1}=\sigma_{j,j+x_{1}}^{\pm }\sigma_{j,j+x_{2}}^{\pm }\ldots \sigma_{j,j+x_{2k+1}}^{\pm }$, where $\sigma_{i}^{-}$ is the inverse braid group element of the generator $\sigma_{j}^{+}=\sigma_{j}$ and the second subscript of $\sigma_{j,j+x_{i}}$ indicates the elementary exchange of *j*-th electron with $\left(j+x_{i}\right)$-th one at the enumeration of electrons on the plane for which every $x_{j}$-th electron is followed by the appropriate next-nearest neighbor in the Wigner lattice and $N/x_{i}$ is the portion of the selected generation of next-nearest neighbors—cf. Appendix B (for $x_{i}=1$, $\sigma_{j,j+x_{i}}=\sigma_{j}$). The more detailed form of the cyclotron subgroup generator is described in Ref. [17].

For the field *B* which satisfies Equation (Equation 9), the degeneracy of Landau levels equals to $N_{0}\left(B\right)=\frac{BSe}{h}$, which gives the filling rate for the corresponding homotopy phase,
(10)$$\nu =\frac{N}{N_{0}\left(B\right)}={\left(\frac{1}{x_{1}}\pm \frac{1}{x_{2}}\pm \cdots \pm \frac{1}{x_{q}}\right)}^{-1}.$$

This is the general hierarchy of FQHE in the LLL.

In [17], we have shown that the energy preference allows for the simplification of Equation (Equation 9) for GaAs via assumption $x_{1}=x_{2}=\ldots x_{q-1}=x$ and $x_{q}=y$, q=2k+1 leading to the following form (and maintaining ± before only last term in (Equation 9)),
(11)$$\frac{S}{N}=\frac{(q-1)h}{eBx}\pm \frac{h}{eBy}.$$

Equation (Equation 11) defines FQHE hierarchy in the lowest Landau level of GaAs,
(12)$$\nu =\frac{N}{N_{0}}=\frac{yx}{(q-1)y\pm x},$$
with q=2k+1 odd integer and y≥x≥1 positive integers 1,2,… (as defined in Appendix B). The hierarchy (Equation 12) reproduces all FQHE filling rations observed experimentally in the LLL of GaAs including so-called enigmatic states impossible to be explained for composite fermion (CF) model, $\nu =\frac{4}{11},\frac{5}{13},\frac{3}{8},\frac{3}{10},\ldots$ [29]. These enigmatic state filling ratios are obtained by the hierarchy (Equation 12), but exclusively for x>1 [17].

This indicates that the phenomenological model of CFs is not effective even in the LLL of GaAs. The CF model assumes in a heuristic manner that to each electron is pinned somehow even number of flux quanta of the auxiliary fictitious magnetic field in order to reproduce the phase shift in the Laughlin function via the Aharonov–Bohm-type phase correction when two such complexes (called composite fermions) interchange their positions on the plane. Moreover, Jain has been suggested [30] that the averaged field of fluxes pinned to electrons may screen the external magnetic field and in the resultant reduced field some higher (spinless) LLs can be completely filled. He confused the *y*-th integer in the formula (Equation 12), which corresponds to the fraction of next-nearest neighbors in the Wigner crystal with *y*-th spinless LL (also integer) and he got the hierarchy $\nu =\frac{1}{(q-1)y\pm 1}$, i.e., the hierarchy (Equation 12) for x=1. Despite a formal agreement in this particular case the Jains’ hierarchy is not able to explain the enigmatic FQHE states as they need X>1. Moreover, this confusion causes the next complications. In the CF model, it is heuristically assumed that the multiparticle wave function for states belonging to the Jain hierarchy can be approximated by wave functions from *y*-th LL in a gaseous spinless system. These states are, however, singular in contrast to states in the LLL and some procedure of the removal of poles occurred necessary. Nevertheless, this artificial procedure (called as the projection onto LLL) [31] is not unambiguously defined and play a role of the variational factor allowing for better energy minimization. This, however, leads to uncontrolled violation of the symmetry of the wave function. This symmetry is properly defined by the cyclotron braid group generators and their scalar unitary representation. Thus, one can conclude that the CF model is wrong and can be at most considered as the pictorial effective illustration for the simplest homotopy phase when nesting of multi-loop cyclotron orbit concerns only nearest neighbors in the Wigner lattice except for a one loop nested with *y*-th fraction of next-nearest neighbors (artificially interpreted in CF model as *y*-th spinless LL). The integer *y* does not mean *y*-th completely filled spinless LL (as assumed in the CF model), but is the value of $x_{q}$ defining the fraction of electrons for consecutive generations of next-nearest neighbors in the Wigner lattice. CFs do not allow for mimicking a more complicated homotopy braid patters involving next-nearest neighbors for the cyclotron commensurability, like for so-called enigmatic FQHE states in the LLL of GaAs [29].

However, for the graphene monolayer, the simplification by Equation (Equation 11) is not admitted due to different envelope functions for Landau level wave functions in comparison to GaAs and the general hierarchy (Equation 10) must be considered. This has been proved experimentally [32,33].

The cyclotron braid generators of cyclotron subgroups are associated with more complicated instances of the homotopy braid commensurability (i.e., with y≥x≥1) as given by (Equation 11) together with the corresponding scalar unitary representations defined in an unambiguous manner for the polynomial part of related multi-particle eigenfunctions. This polynomial must be regular dependent on all particle coordinates, which, together with the requirement to transform itself according to scalar unitary representation of the cyclotron braid group, leads to its unique form for each homotopy phase. The polynomial must be multiplied by nonsingular (as in the LLL any multiparticle wave function must be a holomorphic function without singularities) factor invariant against particle exchanges, thus of the form of an *N*-fold product of single-particle exponents $e^{-|z_{i}{|}^{2}/4l_{B}^{2}}$ ($z_{i}$ is the complex position of *j*-th particle on the plane and $l_{B}=\sqrt{\frac{\hbar }{eB}}$ is the magnetic length). Such a form of the exponential factor is the same as in the gas system and is maintained for an arbitrary state when inter-particle interaction is switched-on. This factor is assumed for a GaAs two-dimensional electron system.

Nevertheless, in graphene, the single-particle Landau states are not of a gaseous form because they are modified by a single-particle crystal field in graphene. This affects the envelope factor invariant against the interchanges of electrons, but the polynomial part that one defined by the homotopy invariant, is the same as in GaAs. The polynomial parts of multi-particle wave functions in the LLL at filling fractions from the general FQHE hierarchy are defined for particular homotopy patterns in an unambiguous manner, as illustrated in [17]. This procedure is exact in contrast to the so-called projection onto the LLL of the wave function for completely filled some higher LL in order to remove singularities in the CF model [31]. The projection onto LLL is not unambiguously defined and breaks the wave function symmetry in an uncontrolled way. This symmetry is precisely defined by the cyclotron braid subgroup generators and their unitary scalar representations. Hence, the trial wave functions in CF model with violated symmetry can be only treated as approximate and optimized in energy gain via the projection recipe variation.

## 4. Topological Correlations and Quantum Entanglement

In homotopy phases defined by appropriate braid groups, i.e., by the full braid group or its cyclotron subgroups, the long range multi-particle correlation of all electrons exists, which causes also long range, throughout the whole system, quantum entanglement of all electrons simultaneously. Multi-particle wave functions corresponding to particular homotopy phases are non-separable functions from the *N*-fold tensor product of single electron Hilbert spaces, $H=H_{1}⊗\cdots ⊗H_{N}$, where $H_{j}$ is the Hilbert space for *j*-th electron. $H_{j}$ is the same space for each *j* though of wave functions with respect to different variables, coordinates of *j*-th electron in the space *M* where electrons are located. In the case when *M* is the 2D plane, coordinates of electrons can be denoted as the complex numbers, $z_{j}=x_{j}+iy_{j}$, i is the imaginary unit. In homotopy phases, all electrons are involved in the multi-particle functions in the same manner. In means that the entanglement is symmetrical and each electron is in a similar mixed state described by the density matrix,
(13)$$\hat{\rho_{j}}=Tr_{1,\ldots .j-1,j+1,\ldots ,N}\left|\Psi \left(z_{1},\ldots ,z_{N}\right)><\Psi \left(z_{1},\ldots ,z_{N}\right)\right|,$$
where |Ψ><Ψ| denotes the density matrix of the pure entangled multi-particle state of all electrons, $Tr_{1,\ldots ,j-1,j+1,\ldots ,N}$ means the trace over the subspace $H_{1}⊗\cdots ⊗H_{j-1}⊗H_{j+1}⊗\cdots ⊗H_{N}$ of the tensor product of Hilbert spaces of all electrons in the system. A typical example of the long-range entangled multi-particle wave function of *N* electrons on the plane is the Laughlin function, $\Psi_{N}\left(z_{1},\ldots ,z_{N}\right)=const\prod_{l>j}^{N}{\left(z_{l}-z_{j}\right)}^{2k+1}e^{-\sum_{j}\frac{|z_{j}{|}^{2}}{4l_{B}^{2}}}$, where $l_{B}=\sqrt{\frac{\hbar }{eB}}$ is the magnetic length. For k=0, the Laughlin function is the Slater function of *N* 2D electrons in the LLL. Apparently, the Laughlin function and the Slater function are nonseparable.

Note that, for a gaseous system without interaction, the full braid group is also properly defined. As gaseous particles do not interact and their distribution is not restricted by any constraints (no Wigner crystal exists in the gas), the cyclotron commensurability loses its sense in this case. The full braid group does not define here the interaction-induced correlations, but only the statistical Pauli correlation (defining bosons, fermions, or anyons in 2D). Despite the Pauli correlation not being induced by any interaction, the multi-particle wave functions both for gaseous fermions and bosons are nonseparable and long-range entangled. This shows that the Pauli correlation leads to the entanglement without any interaction. The role of the interaction here plays the indistinguishability of identical particles. This property has the same features as the binary interaction: (1) needs at least two particles to be considered, (2) concerns all particles in the system in an uniform manner, and (3) disappears for a single particle. These features shared by the indistinguishability condition with the true interparticle interaction are sufficient to induce the long range entanglement. The entanglement is the property of the wave function, which cannot be disentangled by any unitary transformation in the Hilbert space [18].

However, only interacting electrons can create the correlated IQHE state, the topological homotopy state with the commensurability condition (Equation 1) being imposed. For the gas, the states with complete filled Landau levels are not IQHE states. Despite the gas of *N* fermions always exhibiting Pauli correlation and related long-range entanglement, the interaction of 2D electrons at magnetic field is needed for the commensurability condition (Equation 1) or (Equation 9) to be fulfilled in order to create some homotopy topologically protected correlated state (including the states of IQHE and FQHE, no such effects cannot occur in the gas).

In states correlated by braids, i.e., in homotopy phases, none of the electrons are distinguished. Thus, all electrons are simultaneously entangled in a symmetrical manner and such an entanglement over the whole system is called the long range quantum entanglement [18]. This is in contrast to the local quantum entanglement being characteristic to phases of multi-particle systems with local order parameters [18]. The coherent superconducting state is a good example of local binary type entanglement of electrons due to Cooper pairing induced by effective attraction of electrons (mediated by phonons). The specific two-particle interaction causes a local entanglement that reflects the dressing of bare electrons with interaction and leads to eventual quasiparticles. The mass operator and a pole of the retarded Green function, defining a quasiparticle, are the relevant notions in such situations of local entanglement. In the case of long range entanglement, this picture is not useful, and the role of the interaction is different. The correlations expressed by braids are nonlocal of topological character and can be expressed by the commensurability condition (homotopy invariant) exclusively for interacting particles [18].

## 5. Homotopy Braid Group Phases

The full braid group $\pi_{1}\left(F_{N}\left(M\right)\right)$ is the first homotopy group [10,23,24] (called frequently as the fundamental group), which collects disjoint nonhomotopic classes of closed loops in $F_{N}\left(M\right)$, i.e., loops from different classes cannot be transformed one into another by any continuous deformation without cutting. Because points in $F_{N}$, which differ only in particle numbering, are unified (due to indistinguishability of identical particles, which has been accounted for via the division of the multi-particle configuration space by the permutation group), elements from the group $\pi_{1}\left(F_{N}\left(M\right)\right)$ thus describe exchanges of particle positions along trajectories of individual particles which entangle into braids. These braids can be disentangled or not. For $M=R^{3}$ (or for higher dimension of *M*), all braids can be disentangled. For $M=R^{2}$ (or for locally 2D manifolds, like a sphere or torus) braids usually cannot be disentangled [10,23,24]. Therefore, in 3D, the braid groups are always finite permutation groups, i.e., exchanges of particles on 3D manifold are only permutations—renumbering of indexes of particles, whereas, in 2D, trajectories of particle exchanges on the plane are also important, which create bunches usually impossible from being disentangled and with infinite possible structure of different weaves creating nonhomotopic tangles. In 2D, the braid groups are thus infinite (but always countable).

Braid groups have the following properties: (1) In gaseous systems (i.e., for noninteracting particles), no constraints on particle separation are imposed and also no constraints are imposed on linking the braids. (2) When particles mutually interact, then the situation changes significantly. In the case when particles repulse themselves and are deposited on neutralizing jellium, just like electrons in 2D crystalline ion structure treated as the positive jellium, then electrons are uniformly distributed and uniformly separated in classical picture. These electrons create the classical Wigner crystal (if one neglects their kinetic energy, in the framework of classical description at temperature, T=0 K). The triangle Wigner crystal is the lowest energy static classical distribution of repulsing electrons on the jellium at zero temperature. If the magnetic field is switched on, then braids must be built of pieces of cyclotron orbits (no other trajectories exist at the magnetic field presence, let us remind readers that braids are classical trajectories), but, in the 2D case, cyclotron orbits have a finite size (are planar in perpendicular magnetic field without the drift motion), and, hence, braids are also of the finite size. In 3D, cyclotron braids are not limited in size because of the drift motion along the field direction. The cyclotron orbit size in 2D depends on velocity (kinetic energy) of particles. If all particles have the same kinetic energy, like electrons in a LLL, then all cyclotron orbits are of the same size. This size in the LLL is given by $\frac{\Phi_{1}}{B}=\frac{h}{eB}$, where $\Phi_{1}=\frac{h}{e}$ is the magnetic field flux quantum. In higher LLs, the size of cyclotron orbits grows proportionally to the LL energy, and equals $\frac{(2n+1)h}{eB}$, where *n* is the Landau index.

Elementary braids $\sigma_{j}$, the exchanges of neighboring particles without any additional loop, are of the same size as ordinary single-loop cyclotron orbits and cannot match particles on the plane if particles are diluted too much, i.e., when $\frac{h}{eB}<\frac{S}{N}$ (*S* is the surface of 2D sample, *N* is the number of electrons). Hence, for sufficiently large magnetic field *B*, the braids $\sigma_{j}$ cannot be defined. In 3D, no such constraint is imposed because, in 3D, a helical drift along the field direction makes braids arbitrarily long at any field value and such long braids may reach arbitrary distanced particles—in 2D, this does not happen.

The full braid group is generated by the elementary braids $\sigma_{j}$, j=1,…,N−1 being the exchanges of *j*-th and (j+1)-th particles. It is always possible to enumerate particles on the plane that j+1 assigns the nearest neighbor with respect to the *j*-th one, though this numeration is not unequivocal in 2D. Due to indistinguishability of particles, it is enough to assure it for one particle in the Wigner lattice [10]. For 2D interacting electrons and for magnetic field $B>B_{0}$ (where $\frac{S}{N}=\frac{h}{eB_{0}}$), the cyclotron braids $\sigma_{j}$ (of size $\frac{h}{eB}$) are too short to reach neighboring particles. In the braid group, there are, however, multi-loop braids expressed by $\sigma_{j}$ generators $\sigma_{j}^{q}$, which for q=2k+1 (*k*—positive integer) also describe exchanges of particles *j*-th and (j+1)-th, but with additional *k* loops. It has been proved above by application of the B–S rule [17] that such braids have a larger size, $\left(2k+1\right)\frac{h}{eB}$, in the case when they are the simplest paths of particle exchanges. The braids $\sigma_{j}^{2k+1}$ generate the subgroup of the original full braid group, and we have called this subgroup the cyclotron braid subgroup.

Braids do not display any particular dynamics in the system, and they can be modified only by topological constraints. In a graphene monolayer, which is a 2D system, braids in the magnetic field presence will be the same as in GaAs 2DES despite different single-particle energy levels. In graphene monolayer LL, energies have the form, $E_{n}=\sqrt{n}\frac{2\hbar v_{F}}{l_{B}}$ (where $v_{F}$ is the Fermi velocity, $l_{B}$ is the magnetic length) [7], whereas in GaAs 2DES $E_{n}=\hbar \omega_{B}\left(n+\frac{1}{2}\right)$, $\omega_{B}=\frac{eB}{m}$, *n* is the Landau index—cf. Appendix A. However, the cyclotron orbit size is governed by the bare kinetic energy and in graphene in the LLL, it is the same as in gaseous system, $\frac{h}{eB}$. Thus, Equation (Equation 9), defining patterns of the commensurability, holds also in the graphene-monolayer (the same is also the degeneracy of each LL subband, $N_{0}=\frac{BSe}{h}$). In the graphene-monolayer, the number of subbands in each LL is, however, different in comparison to GaAs—in the graphene monolayer, this number is four, due to the spin-valley structure [7] (in GaAs 2DES, only two spin subbands occur). Due to a Barry phase shift in the graphene-monolayer, the zero energy at the Dirac cone tops is also zero LL energy [7]. These all are, however, only corrections of the single particle energy spectrum induced by the electric-type crystal field not perturbing neither the bare kinetic energy nor the braid homotopy invariants. In bilayer-graphene, the situation is similar, though the LLL is 8-fold split (SU(4) × 2) due to an additional degeneracy of Landau oscillatory states with indices n=0 and n=1 in graphene bilayer [34]. Not perfect 2D structure in the bilayer-graphene is, however, a topological factor which significantly modifies the braid commensurability conditions in this material resulting in different homotopy patterns and topological invariants [28].

In bilayer graphene, inter-layer hopping of electrons is admitted. Hence, loops of cyclotron orbit can also hop between layers. Each layer has its own surface and separately contributes to the total flux of the magnetic field *B* passing the bilayer graphene (like in 3D case). Thus, for a multi-loop structure of the cyclotron orbit, loops can be arbitrarily distributed among layers and individually adjusted to nearest or next-nearest neighbors in electron Wigner lattice in each layer, which perturbs the commensurability condition [28,35].

By application of the vertical voltage, one can restore the monolayer homotopy in bilayer graphene. A vertical voltage can block electron hopping in one direction, which completely precludes hopping of loops (braid loops are closed, thus must return back if hops)—FQHE hierarchy in the bilayer graphene exposed to the vertical voltage becomes that one as in the monolayer graphene. This effect has been observed experimentally [36].

In higher LLs in every Hall material, with the Landau index n>1, the quantized bare kinetic energy is (2n+1)-times greater than the energy in the LLL (because $E_{n}=\hbar \omega_{B}\left(n+\frac{1}{2}\right)$, where $\omega_{B}=\frac{eB}{m}$). Therefore, cyclotron orbits in higher LLs are (2n+1)-times larger in comparison to the LLL. The metrics of braids in higher LLs are thus also (2n+1)-times larger than the metrics of $\sigma_{j}$ in the LLL. It means that, in higher LLs, the loop-less braids $\sigma_{j}$ (corresponding to the ordinary single-loop cyclotron orbits) can match particles more separated than in the LLL. This completely changes the commensurability condition (Equation 9) and explains why the experimentally observed hierarchy of FQHE in higher LLs is different than that in the lowest one. The cyclotron commensurability with electron separation in higher LLs has been analyzed in [37,38] for GaAs consistent with the experiment [39]. In particular, in GaAs 2DES, the experimentally observed FQHE states at $\nu =\frac{7}{3},\;\frac{8}{3}$ and $\nu =\frac{10}{3},\;\frac{11}{3}$ in two spin subbands for n=1 (and similar states at fillings with the denominator 5 for subbands with n=2)—cf. Figure 4, are single-loop homotopy phases in contrast to similar fractional fillings of the LLL (fractions $\nu =\frac{1}{3},\;\frac{2}{3}$ and $\nu =\frac{4}{3},\;\frac{5}{3}$ in two spin subbands with n=0) at which the multi-loop FQHE occur [37]. This important observation indicates that FQHE in higher LLs is not of a CF type because the latter can be utilized as the phenomenological picture for multi-loop braids homotopy phases in the LLL and are useless for single-loop homotopy phases in higher LLs [37].

The same holds in graphene. The cyclotron commensurability with electron separation in graphene monolayer and bilayer has been successfully applied to explain an unconventional FQHE in these Hall materials [17,28].

Finally, let us address the homotopy approach to electrons in topological Chern insulators, where the Hall physics are reproduced but without a uniform magnetic field and without LLs. The role of magnetic field takes the Berry field [9] with similar flux quantization. The Berry field flux quantum corresponds here to single circulation along the circumference of the planar elementary cell in crystalline structure. The degeneracy of Landau levels is substituted in Chern topological insulators by the number of states in an extremely flattened band which is equal to the number of crystalline nodes in the sample, $n_{0}$. Thus, the filling rate is counted per crystalline node, $\nu =\frac{N}{n_{0}}$. The typical FQHE hierarchy found in numerical simulations of fractional Chern topological insulators at the same as for FQHE fractional fillings $\nu =\frac{N}{n_{0}}$ of a flattened band counted by the filling of 2D crystal lattice nodes [40] strongly emphasizes the universal character of FQHE in various 2D systems and reflects the same homotopy classification of electron trajectories (the same homotopy invariants) as in the ordinary Hall systems.

The discussion presented above and the review of homotopy correlation phases in various materials are summarized in Table 1. In the listing in the table, the metrics imposed on braids by the cyclotron effect is highlighted in 2D geometry, and not in 3D. The metrics can appear in 2D electron systems regardless of the electron-interaction (i.e., both for interacting electrons and noninteracting gaseous models) as the response to the action of the perpendicular quantizing magnetic field (in conventional Hall systems) as well as of the Berry field (in the case of Chern topological insulators). The role of the interaction of electrons in arrangement of IQHE and FQHE strongly correlated states is visible due to existence of the Wigner lattice exclusively for repulsing electrons. The commensurability patterns being homotopy invariants are listed both for the LLL in GaAs, graphene monolayer and bilayer, and Chern topological insulators. The specific non-CF-type FQHE hierarchy in all of these materials is addressed to single-loop homotopy phases in higher LLs and in graphene bilayer both in LLL and in its higher LLs taking into account jumps of cyclotron loops between layers. The presented homotopy classification found a perfect consistency with current experimental observations. The particularities of the hierarchy and comparison with experiment are considered in details in references indicated in Table 1.

## 6. Conclusions

By application of homotopy methods, the classification of 2D interacting multi-electron systems in the magnetic field (or Berry field) is given (and summarized in Table 1). The structure of correlations responsible for IQHE and FQHE is recognized consistent with experimental observations in 2DES of GaAs and in graphene monolayer and bilayer. It is shown that the homotopy invariants related to the effective flux quanta for various correlation patterns are immune to the variation of interaction strength (though must be nonzero) and local crystal field, and thus the universal hierarchy of FQHE is repeated in all 2D charged Hall systems in a perpendicular quantizing magnetic field (like in GaAs 2DES or graphene) as well as in a quantizing Berry field in fractional topological Chern insulators. Homotopy invariants are also resistant to local thermal chaos, and corresponding homotopy phases are visible at the same filling rates unless thermal energy kT exceeds the energy gain due to topological correlations (in correlations in higher LLs, they take part in lowering portions of electrons, thus the energy gain is gradually reducing and experimental observation of the homotopy features in these levels needs lower temperatures). The unitary representations of cyclotron braid subgroups for particular homotopy phases allow for the reconstruction of the multi-particle wave functions of these phases and the assessment of energy consistent with Laughlin function form, exact diagonalization of interaction in small models and experimental observations of stable states at various filling rates of LLs. Despite the degeneracy of LLs being lifted by the electron interaction, the homotopy invariants still allow for identifying different topological phases with long range correlation and long range quantum entanglement at topologically protected filling rates conventionally expressed in terms of the degeneracy of single-particle LLs being the same as the number of band-states in interacting multi-electron systems. The stability of different homotopy patterns is a matter of the envelope part of the corresponding multi-particle wave function also immune to the electron interaction but dependent on a crystal field in various materials (as observed in graphene in distinction to observations in GaAs). The interaction between electrons is, however, essential for the definition of topological correlations (in gaseous systems, they cannot occur) which are governed by homotopy invariants possible to be defined only in interacting 2D multi-electron systems, which have been explicitly demonstrated.

## Figures and Tables

**Figure 1 materials-14-01650-f001:**
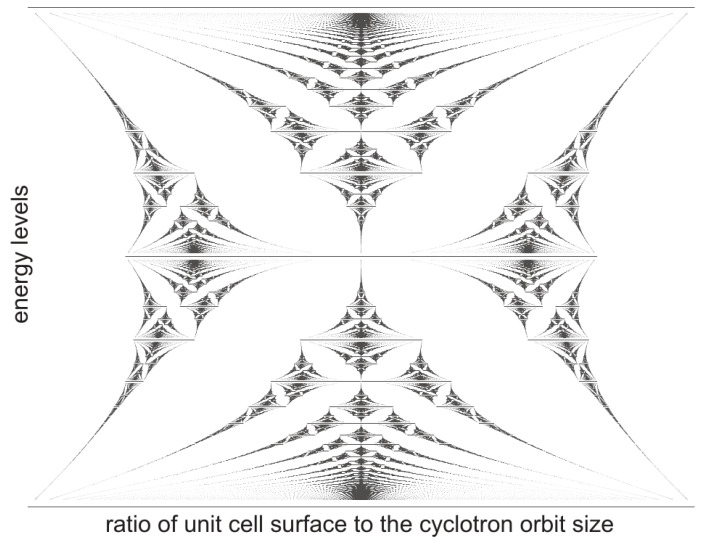
Rendering of Hofstadter butterfly—a fine structure of LLs for the single electron in the periodic square type 2D external potential is shown as the function of the commensurability factor of the cyclotron orbit size with the unit crystal cell. The energy levels are indicated versus the magnetic field flux through the elementary cell, $a^{2}B$, expressed in units of the elementary flux $\frac{h}{e}$, i.e., versus the ratio of the unit cell surface $a^{2}$ to the electron cyclotron orbit size $\frac{h}{eB}$ [19,20].

**Figure 2 materials-14-01650-f002:**
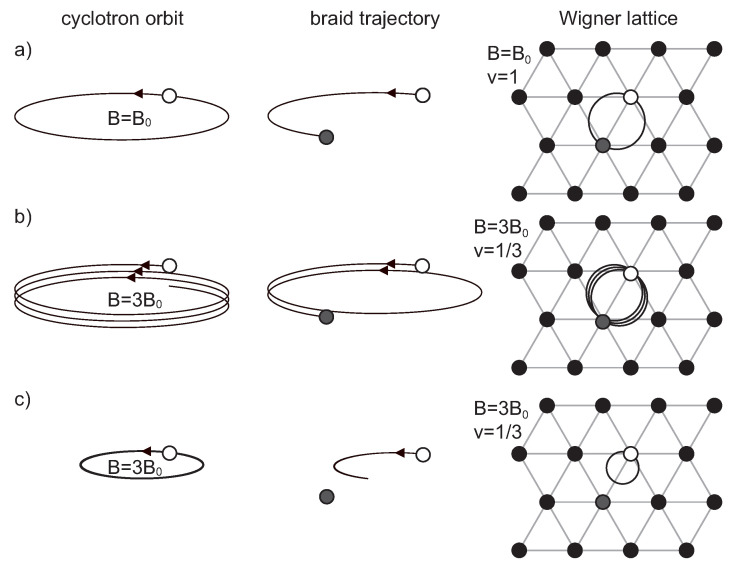
(**a**) left panel—a single-loop cyclotron orbit is schematically presented together with the corresponding braid generator $\sigma_{j}$ of the full braid group (central panel)— its metrics perfectly fits at ν=1 to the separation of interacting electrons in the Wigner lattice on the plane (right panel). (**b**) left panel—three-loop cyclotron orbit for a three-times larger magnetic field and the related generator of cyclotron braid group $\sigma_{j}^{3}$ (central panel)—its metrics also perfectly fit at $\nu =\frac{1}{3}$ to the electron separation in the Wigner lattice (right panel); (**c**) a visualization that the small single-loop cyclotron orbits for the field three times larger than in the case of (**a**) i.e., at at $\nu =\frac{1}{3}$ preclude the definition of braids $\sigma_{j}$ (such braids cannot reach even the closest electrons in the Wigner lattice (right panel)).

**Figure 3 materials-14-01650-f003:**
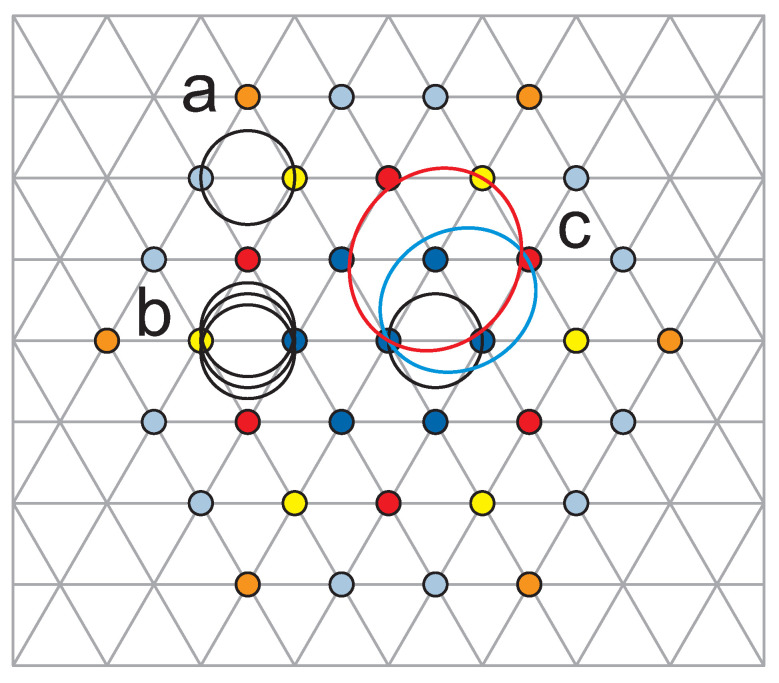
Schematic presentation of exemplary commensurability patterns acc. to Equation (Equation 9) for single-loop cyclotron orbit and ν=1 (**a**), three-loop cyclotron orbit with all loops nested with nearest neighboring electrons and $\nu =\frac{1}{3}$, i.e., $x_{1}=x_{2}=x_{3}=1$, q=3 in Equation (Equation 9) (**b**) and three-loop cyclotron orbit with loops nested with next-nearest electrons, i.e., for $x_{1}=1$, $x_{2}=3$ and $x_{3}=4$, q=3 in Equation (Equation 9), for which $\nu ={\left(1/1+1/3+1/4\right)}^{-1}=\frac{12}{19}$ (cf. Figure A2 for the explanation of values for $x_{i}$ displaying fractions of succeeding generations of next-nearest neighbors in the Wigner lattice) (**c**).

**Figure 4 materials-14-01650-f004:**
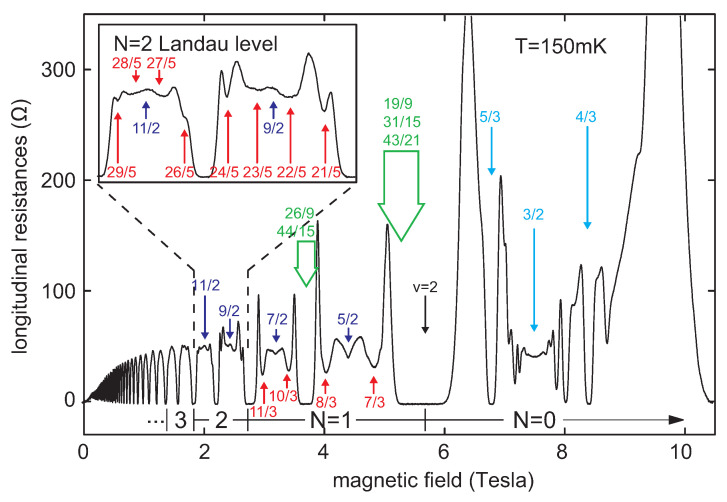
Visible in experiment [39] single-loop FQHE states for Landau index n=1 i n=2 (*N* in the figure) (in GaAs)—indicated in red color; blue ones—4/3, 5/3 are three-loop FQHE in the lowest Landau level at n=0; ν=5/2,7/2,9/2,11/2 are paired states, but at 3/2 is the Hall metal; the green color marks a few FQHE from multi-loop series pushed toward band edges at n>0 and obscured by IQHE-*reentrant* in the vicinity of integer fillings [17,37].

**Table 1 materials-14-01650-t001:** Classification of multi-electron homotopy phases [m(b)GN—monolayer(bilayer)-graphene, ChTI—Chern topological insulator].

Dim.	System	Braid Metrics and Nesting Type	Homotopy Phases
3D	gas or interacting electrons	no braid metrics	no homotopy correlation phases except for Pauli correlations
2D	gas	no cyclotron braid nesting, no Wigner lattice	no homotopy correlation phases except for Pauli correlations
2D	interacting electrons	single-loop cyclotron braid nesting in Wigner lattice	homotopy phases of IQHE, $\nu =\frac{N}{N_{0}}=1$; FQHE in higher LLs, ν=2+(1[i])+1/3(2/3),4[6]+(1[i])+1/5(...,4/5) (GaAs), ([...]mGN, i=1,2,3) [17,28,41]
2D	interacting electrons	multi-loop cyclotron braid nesting in Wigner lattice	homotopy phases of FQHE, $\nu =\frac{xy}{(q-1)y\pm x}$ (GaAs) [17] $\nu ={\left(\frac{1}{x_{1}}\pm \cdots \pm \frac{1}{x_{q}}\right)}^{-1}$ (mGN) [33]
bGN 2D-2D	interacting electrons	multi-loop cyclotron braid nesting in double Wigner lattice	homotopy phases of FQHE, interlayer distribution of loops and interlayer flux leakage [28]
ChTI 2D	gas	no Berry braid [9] nesting, no Wigner lattice	no homotopy correlation phases except for Pauli correlations
ChTI 2D	interacting electrons	single-loop Berry braid nesting in Wigner lattice	homotopy phases of IChTI (integer ChTI), $\nu =\frac{N}{n_{0}}=1$, $n_{0}$ number of nodes
ChTI 2D	interacting electrons	multi-loop Berry braid nesting in Wigner lattice	homotopy phases of FTChI (fractional ChTI), $\nu =\frac{N}{n_{0}}=\left(\frac{1}{x_{1}}\pm \cdots \pm \frac{1}{x_{q}}\right)$

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
