# Peer review of "Topological Classification of Correlations in 2D Electron Systems in Magnetic or Berry Fields"

_materials, 2021, doi:10.3390/ma14071650_

Round 1
Reviewer 1 Report
This paper gives an extended overview, based on the author's own work, on the topological classification of 2D electron systems with and without interaction that lead to the integer and fractional Quantum Hall effect, topological insulators, Weyl metals etc. due to their intriguing correlations.
The homotopy phases turn out to be topolocially protected, but only the finite interaction between electrons allows the definition of homotopy invariants
The general topic of the paper is timely, and the general approach outlined in the paper is scientificly sound. Apart from some small problems with English, the paper is well written for the reader who has already some background within the framework of this type of theory.
Therefore, a more extended introduction would be helpful that starts more with a motivation and with an explanation of the general approach, including the mathematical terms used here starting with the first sentence of the introduction.
Author Response
Dear Referee,
thank you very much for a positive opinion. According to your suggestion we have improved the resubmission.
The developed general description is added in the Introduction (marked in color in the corrected pdf), with new 10 references.
The polishing of English language in the text has been done by the author with the help of a native speaker.
In this way we have addressed to all your suggestions and hope that we have made it in a satisfactory manner.
Sincerely yours,
J. E. Jacak
Reviewer 2 Report
The manuscript presents a topological classification of many-electron states of 2D systems in a perpendicular magnetic field. This is an extension of the work of others in the absence of a magnetic field. Both the effect of electron correlation and of an external magnetic field are critical extensions of the simple band theory of topological materials. Hence, this work should be of wide significance. It would be necessary to significantly improve the English.
Author Response
Dear Referee,
Thank you very much for a positive opinion.
According to your suggestion an extensive editing of the text has been performed by the author with the help of a native speaker.
Some developments of the text enhancing clarity of the presentation of results and the introduction are added and marked in color in the improved resubmitted PDF file.
We hope that the corrections are done in a satisfactory way.
Sincerely yours
J. E. Jacak
Reviewer 3 Report
This manuscript is, as it claims to be, on topological classification of 2D correlated electronic states in the presence of strong magnetic field. The main point of this manuscript seems to strongly relying on the concept of cyclotron braid group, which is a novel suggestion of the author (PRA 97, 012108). It looks indeed an interesting result, enough to warrant its publication in a scientific journal like Materials, although the author could have chosen a bit more theoretical journals such as PRA.
Problem is... most of the volume of this paper seems to be devoted to just rephrasing what was already published, especially Ref. 10 in this manuscript (J. Jacak, PRA 97, 012108 (2018)). The only new result of the paper, compared to the previous works, is Table I, which seems indeed relevant to the title of the manuscript. Surprisingly, the table is never mentioned in the main text, just once in the summary section at the end of the paper. Furthermore, I cannot see in any place how the classification was made or what homotopy group argument the author is employing. The author is repeatedly trying to explain difference between strongly correlated Wigner crystal and noninteracting gas, but it doesn't well connect to the classification scheme that is promised in the title and abstract.
Therefore, I cannot recommend the publication of this manuscript in Materials as it is now. The author should add more new and meaningful information here compared to previous studies, or at least the way of presentation needs to be improved enough for the new content to be delivered to readers.
Author Response
Dear Referee,
Thank you very much for your recommendations and critique. We have improved the resubmission according to all your suggestions,
We described in the Introduction the novelty of the presentation and consideration with respect to our former publication [10] (now [17]) in PRA. In fact the former discussion in [10] has been concentrated on the Feynman path development onto multiply-connected configuration space of 2D electrons exposed to quantizing magnetic field (as is clearly stated in the title of [10]). In the present submission we look at the problem from a bit different perspective -- the wave functions with topology invariants and metrics of braids acquired due to quantization of the magnetic or Berry field fluxes. The path integral is even not mentioned in the present submission taking into account that the correlations imposed on electrons by a particular homotopy invariant causes an appropriate symmetry of the multiparticle wave function independently of the path integral formulation (as shown previously by Sudarshan and Imbo). We demonstrate it (in the corrected resubmission) on a transparent example of two electrons fulfilling two different homotopy conditions (It has not been presented in [10], similarly as the precise definition of factors x_i in the topological invariant (9) in close and explicit relation to next-nearest neighbor generations in the Wigner crystal, originally derived in the present submission). This is of central importance for the commensurability condition and the final general fractional hierarchy. This has never been published previously. To show it more distinctly we added some comments to the text highlighted in the color (red) -- in the Introduction and in the Appendix C (including two new figures). We emphasized also in the present submission (not mentioned in [10]) the fundamental difference between homotopy phases without any local order parameter in comparison to conventional phases with broken some symmetry and with local order parameters. This difference consists in long range quantum entanglement of all electrons simultaneously in the system in a topological phase in contrary to conventional phases with only local binary quantum entanglement of interacting electrons. The novelty concerns also the Table 1 summarizing the homotopy classification in various systems. We have added (according to your suggestion) an extended paragraph explaining details referred to the Table 1 -- also highlighted in color (red). The intensive proofreading has been additionally performed.
We hope that we have improved the resubmission in a satisfactory way.
Sincerely yours,
J. E. Jacak
Reviewer 4 Report
This paper theoretically investigates a homotopy classification of the topologically protected correlations responsible for integer and fractional Hall effects in 2D electron systems in the presence of perpendicular magnetic fields or Berry fields.
The main achievements of this paper are as follows:
- The author provides a homotopy classification of correlation patterns of 2D interacting electron systems with vertical strong magnetic fields, utilizing the commensurability condition between electronic cyclotron motion and 2D periodic lattice.
- The developed method of classification is applied to quantum Hall states observed in 2D semiconductors, graphene monolayers/bilayers, and Chern topological insulators.
The subject on quantum Hall physics picked up in this article has a long history and still attracts the interest of broad readership. This article is suitable for "Materials", while there remain several improvable points listed below.
- In my understanding, I think that the core part of this paper is the flow of logic from Equations (8) to (10). However, the explanation from Equation (8) to Equation (9) has not a few sensuous or graphical parts. So, if there is any leap in logic at least in the present stage of the study, I'd like to recommend the author to comment honestly on that. I believe that such comments will never reduce the value of this discussion derived from a unique heuristic approach. Rather, it will inspire readers to gain a deeper understanding of this article.
- Although the explanations in the text are understandable for non-specialists, many readers would be unnecessarily confused by the situation where the same symbols are used for different objects, e.g. "h" for Planck constant and the height of a unit triangle, "i" for the indices of i-th electron and i-th loop. I'd like to ask the author to be considerate of the readers (including me) who are not familiar with the way of the discussion presented in this article yet. Any improvement on this issue would be appreciated.
I recommend the publication of this paper after the revisions on the points listed above and on typos in the version of the manuscript submitted for peer-review.
Author Response
Dear Referee,
Thank you very much for the recommendations and the overall assessment of our submission. We fully agree with your critique and suggestions to improve the presentation. According to these suggestions, we added the explanation of the derivation of Eqs (8)-(10) (especially the step between (8) and (9)) in more detail -- highlighted in red color ) in the improved resubmitted PDF file. Moreover, we added also a new illustration (Fig. 3 in the present verified version) to better clarify the derivation of the central result. According to your suggestion we have changed some symbols used previously repeatable in distinct meanings. In Fig. A3 we have changed h -- the height in the triangle in Wigner lattice to d, in order to distinguish of conventional signing of Planck constant. We have also changed the numbering of electrons to j letter in order to distinguish it from numbering of x_i referred to next-nearest neighbor fractions in Wigner lattice.
Moreover, the extensive proof-reading has been performed. Some corrections are indicated with color.
We hope that we have improved the resubmission in a satisfactory manner.
Sincerely yours,
J. E. Jacak
Reviewer 5 Report
In the present paper the author consider a classification of strongly correlated two-dimensional electron systems based on homotopy invariants. He is also able to connect this results with experimental evidences on two-dimensional electron gas, monolayer and bilayer graphene and topological insulators.
In my opinion the paper is well written, scientifically sound and very interesting. Indeed, I think it could shed new light on the classification of correlated states of matter, a relevant and still quite unexplored subjec. According to this I recommend the publication of the paper on Materials in the present form.
As a minor comment I suggest the author to do an overall spell check of the manuscript.
Author Response
Dear Referee,
Thank you very much for an excellent opinion and recommendation.
According to your suggestion the proofreading of the text has been done. Some corrections are marked in color.
We are obliged for your opinion.
Sincerely yours,
J. E. Jacak
Round 2
Reviewer 3 Report
The author has addressed criticisms and concerns raised in my previous report. I now recommend the publication of this manuscript in Materials.